# Analysing Normative Influences on the Prevalence of Female Genital Mutilation/Cutting among 0–14 Years Old Girls in Senegal: A Spatial Bayesian Hierarchical Regression Approach

**DOI:** 10.3390/ijerph18073822

**Published:** 2021-04-06

**Authors:** Ngianga-Bakwin Kandala, Chibuzor Christopher Nnanatu, Glory Atilola, Paul Komba, Lubanzadio Mavatikua, Zhuzhi Moore, Dennis Matanda

**Affiliations:** 1Division of Health Sciences, Warwick Medical School, University of Warwick, Coventry CV4 7AL, UK; 2Division of Epidemiology and Biostatistics, School of Public Health, University of the Witwatersrand, Johannesburg 2193, South Africa; 3Department of Mathematics, Physics & Electrical Engineering (MPEE), Northumbria University, Newcastle NE1 8ST, UK; chibuzor.nnanatu@northumbria.ac.uk (C.C.N.); glory.atilola@northumbria.ac.uk (G.A.); paul.komba@northumbria.ac.uk (P.K.); mavatikua3@gmail.com (L.M.); 4Independent Consultant, Vienna, VA 22182, USA; zhuzhimoore@gmail.com; 5Population Council, Avenue 5, 3rd Floor, Rose Avenue, Nairobi, Kenya; dmatanda@popcouncil.org

**Keywords:** FGM/C, spatial analysis, bayesian hierarchical modelling, social norms

## Abstract

Background: Female genital mutilation/cutting (FGM/C) is a harmful traditional practice affecting the health and rights of women and girls. This has raised global attention on the implementation of strategies to eliminate the practice in conformity with the Sustainable Development Goals (SDGs). A recent study on the trends of FGM/C among Senegalese women (aged 15–49) which examined how individual- and community-level factors affected the practice, found significant regional variations in the practice. However, the dynamics of the practice among girls (0–14 years old) is not fully understood. This paper attempts to fill this knowledge gap by investigating normative influences in the persistence of the practice among Senegalese girls, identify and map ‘hotspots’. Methods: We do so by using a class of Bayesian hierarchical geospatial modelling approach implemented in R statistical software (R Foundation for Statistical Computing, Vienna, Austria) using R2BayesX package. We employed Markov Chain Monte Carlo (MCMC) techniques for full Bayesian inference, while model fit and complexity assessment utilised deviance information criterion (DIC). Results: We found that a girl’s probability of cutting was higher if her mother was cut, supported FGM/C continuation or believed that the practice was a religious obligation. In addition, living in rural areas and being born to a mother from Diola, Mandingue, Soninke or Poular ethnic group increased a girl’s likelihood of being cut. The hotspots identified included Matam, Tambacounda and Kolda regions. Conclusions: Our findings offer a clearer picture of the dynamics of FGM/C practice among Senegalese girls and prove useful in informing evidence-based intervention policies designed to achieve the abandonment of the practice in Senegal.

## 1. Introduction

The World Health Organisation (WHO, Geneva, Switzerland) defines female genital mutilation/cutting (FGM/C) as a procedure encompassing partial or total removal of external female genitalia on non-medical grounds. The practice has been classified into four categories: Type I (or clitoridectomy) involves partial or total removal of the clitoris or only the prepuce in rare cases. Type II (or excision) is the partial or total removal of the clitoris and the labia minora and sometimes along with the labia majora. Type III (or infibulation) is the cutting and repositioning of the labia minora or majora to create a covering seal sometimes through stitching in order to narrow the vaginal opening. Type IV includes all other harmful alterations or injury to the female genitalia (not classified under Types I–III) such as pricking, piercing, incising, scraping and cauterization [1]. Several reasons for practising FGM/C which include social acceptance, cleanliness, marriage prospects, marital fidelity, preservation of virginity, amongst others have been adduced by the adherents of the practice [2]. 

Despite increased recognition of its harmful health outcomes [1,2,3,4,5,6,7,8], FGM/C is still practiced in 30 countries in Africa, Middle East and Asia. Given its negative outcomes and human rights concerns for women and girls around the world, FGM/C has become a long-term focus of concerted efforts to ensure that the practice is eliminated, mainly in order to meet the Sustainable Development Goals (SDGs) [9,10].

Statistical evidence has shown a wide range of inter-cluster and intra-cluster variations in the practice of FGM/C among women and girls in Senegal, despite moderate to low national prevalence estimates over time. For example, 2005 Senegal Demographic and Health Survey (SDHS) showed that the national prevalence of FGM/C among women aged 15–49 years in Senegal was 28% while prevalence was up to 94% among women who lived in Kolda region in the same year [11,12,13]. In addition, in a study by Kandala and Komba using evidence from 2010 SDHS, they found wide variations within stratum and between strata, where prevalence among women aged 15–49 who lived in rural areas was reported to be 43.1% against the 35.5% of urban women [14]. These non-homogeneous variations were supported by more recent studies by Kandala and Shell-Duncan [15] and Matanda et al. [16]. Specifically, Kandala and Shell-Duncan identified ethnicity, religion, type of place of residence (urban–rural) and region of residence as key factors driving the prevalence of FGM/C among women aged 15–49 using evidence from 2005 and 2010 SDHS [15]. They found that the risk of FGM/C was higher among younger women (15–20 years), among Mandingue, Soninke and Diola women, among rural areas dwelling women, and among women who lived in Matam region. A descriptive report by Matanda et al. reported unadjusted variations in the practice among Senegalese girls aged 0–14 years in addition to the variations found among women aged 15–49 years [16]. Evidence from the successive SDHS data from the 2005 to 2017 DHS (Demographic and Health Survey) showed an insignificant unadjusted decline in the practice between 2015 and 2017. 

In Senegal, varieties of interventions ranging from government policies and support as well as anti-FGM/C laws, education, re-orientation to empowerment programmes are underway. For example, the Senegal Penal Code of 1999 criminalised FGM/C with offenders upon conviction, serving six months to five years in jail, or hard labour for life where the practice results in death [12]. The Penal Code of 1999 serves to deter adherents from indulging in the practice with the aim of a total eradication of the practice. In addition, the popular Tostan project which works in collaboration with other partners such as the Orchid Project, aims to empower communities through its Community Empowerment Programme (CEP) by providing informal education to the affected communities on varieties of topics including health and human rights (www.tostan.org, accessed on 7 August 2019. Other interveners such as the Grandmother Project (GMP) also known as Girl’s Holistic Development (GHD) actively engages older women (grandmothers) who are often the revered elders in the communities, supports intergenerational communities and promotes harmless cultural values, while harmful practices such as FGM/C and child marriage are discussed and discouraged. The objective of GMP is to re-orientate and educate the affected communities to have increased awareness of the dangers of harmful traditional practices and ultimately jettison them whilst upholding positive cultural values [12]. However, the findings from [14,15,16] raised the important question on why FGM/C is still sustained in Senegal despite the huge investment in intervention. 

One body of the FGM literature suggests that social norms are the key drivers of the practice [17,18,19]. Social norms theory posits that patterns of behaviour are influenced by unofficial social rules. These rules are learned through social interactions with reference group in the community, suggesting that FGM/C persists through these interactions and reciprocal expectations of those in the reference group [17].

On this view, actions taken by individuals are viewed as governed not only but their own preferences and attitudes, but also by the influences of others’ expectations and pressure to conform.

Gerald Mackie hypothesised that FGM/C was a social norm which spread and became locked in place by interdependent expectations regarding marriageability [18]. In this way, FGM/C provided an advantage as cut women are believed to maintain marital fidelity and could be regarded as a prerequisite for marriage. Another interpretation of the social convention theory suggests that FGM/C is a requirement not only for enhancing women’s chances of marriage but also for being socially accepted and included in social networks of cut women [19]. On this account, uncut women often pay the high price of foregoing marriage, legitimate childbearing or even become ostracized.

According to DHS-MICS 2010–2011 report, in Senegal, FGM/C is practised on various ethno-cultural and socio-cultural grounds. For instance, the Mandingue ethnic group, who are mainly Muslims, view FGM/C as part of their religion [20], and like the Poular and Soninke ethnic groups, mainly practise the Type III (infibulation) as a ritual in order to preserve the virginity of their daughters 94% of whom are cut before their fourth birthday [13].

Following from the above, there is evidence that FGM/C in Senegal is deep-rooted in socio-cultural norms across ethnic and religious lines and it becomes imperative to deeply assess the roles of these norms in the persistence of the practice in Senegal.

To address this, we used a class of advanced statistical techniques implemented in a coherent Bayesian hierarchical modelling framework to assess both unadjusted and adjusted influence of social norms as well as the influence of an individual’s geographical location on a girl’s likelihood of experiencing the practice. In particular, we used Bayesian hierarchical geo-additive spatial logistic regression models to model and map hotspots where the practice is still rife largely due to the influence of social norms. We note that this Bayesian hierarchical modelling approach, popular in disease mapping studies [21,22], has also been applied recently in FGM/C related studies [15,22,23,24,25]. The approach employed in this paper is an extension of that used in Kandala and Shell-Duncan [15], which studied Senegalese women aged 15–49 years. For our purposes, we focused on Senegalese girls aged 0–14 years. The approach was further extended to account for the effect of time and space-time interaction by including terms on time and space as well as their interaction in the model.

In the light of the above, we summarise the aims of this present study as follows:(1)Assessment of the trends of FGM/C prevalence among Senegalese girls aged 0–14 years focusing on the roles of social norms in the persistence of the practice.(2)Assessment of the unobserved effects of geographical location as well as time and space-time interaction on the likelihood of FGM/C among 0–14 years old Senegalese girls.

We structure the remainder of this paper as follows. In Section 2, we describe the methods including data and variables employed. A brief introduction of the statistical approach used here is given in Section 3. In Section 4, we detail the results obtained from the various analyses conducted for our purposes including descriptive and the Bayesian spatial and spatio-temporal model analyses. In Section 5, we discuss the significance of our findings on the public health and wellbeing of Senegalese women and girls, and we conclude by highlighting the implications of the key findings in Section 6. 

## 2. Materials and Methods

### 2.1. Data

We analysed data collected in the nationally representative Senegalese Demographic and Health Surveys (SDHS) from 2005 to 2017. Two sets of data—the 2017 SDHS and a pooled data from 2010 SDHS to 2017 SDHS were used. With the exception of the 2005 SDHS, the other surveys included information on all the fourteen regions of Senegal. These regions are Dakar, Diourbel, Fatick, Kaffrine, Kaolack, Kedougou, Kolda, Louga, Matam, Saint Louis, Sedhiou, Tambacounda, Thies and Ziguinchor (see Figure 1). The main inhabitants of the Matam region are the Poular ethnic group. Saint Louis region is inhabited mainly by the three ethnic groups of Poular, Wolof and Toucoulour. The inhabitants of Tambacounda region include Poular, Mandigue, Toucoulour, Soninke, Wolof and other minority groups. In addition, Kolda region is inhabited by Poular, Soninke and Mandigue ethnic groups. The region of Sedhiou inhabits mainly the Mandigues along with Toucolour and other ethnic minorities. Finally, Diolas are the major ethnic groups found in the Ziguinchor region.

The FGM/C module in the survey included three key sections: (1) whether the woman underwent FGM/C or not, and details about the event, (2) whether the respondent’s daughter(s) underwent FGM/C or not, and details about that event and (3) the woman’s opinion about the continuation of the practice. Questions on FGM/C were asked to women who received the full survey including questions on whether they had ever heard of “female circumcision”. Respondents who had not heard of female circumcision were asked if they had ever heard of a practice in which a girl has part of her genitals cut.

In 2005, 2010–2011, 2015 and 2017 SDHS, 377, 391, 214 and 400 clusters were selected, respectively. These clusters were sampled and all the women in the selected households from the random clusters are then interviewed. Data for girls aged 0–14 years are then collected from such women who had at least one living daughter aged 0–14 years. The sampling design, organisation, sample size, questionnaires, and implementation used for each survey are described in the respective survey reports (https://www.dhsprogram.com/, accessed on 3 May 2018).

Table 1 shows the sample sizes of girls aged 0–14 years as well as the national prevalence across the survey years.

### 2.2. Outcome Variable

The only outcome variable used in this study was the FGM/C status of a girl. The outcome measure was defined as a binary variable and takes the value of zero (0) if a girl is uncut and one (1) if a girl has been circumcised.

### 2.3. Exposure Variables

We used individual- and community-level characteristics along with the variable that identifies the geographical location of a girl and her mother. In addition, we used mother’s location (urban vs. rural, region of residence) to assess the influence of social interaction between two neighbouring regions. Variables used as proxies for social norms included individual-level (a woman’s FGM/C status, a woman’s support for FGM/C continuation, a woman’s belief about FGM/C being a religious dictate) and community-level (proportion of cut women in the community) characteristics. Other exposure variables included mother’s age, level of education, household wealth and religion.

## 3. Statistical Analysis

### 3.1. Bivariate Data Analysis

To undertake weighted bivariate analyses on the datasets, we used Stata version 14 (Sata Corp 2009. https://www.stata.com/, accessed on 3 May 2018) via the *svy* command. This was necessary in order to test the pairwise strength of association between the covariates and the response variable [26].

### 3.2. Bayesian Hierarchical Spatial and Space-Time Modelling

The cluster sampling used by the SDHS to select the respondents implies that it is wrong to assume that observations are independent or uncorrelated. In this respect, we deemed it inappropriate to use standard statistical methods given that such methods are only valid when observational units are independent. Thus, there was a need to employ an advanced statistical approach that accounts for correlated responses and controls for other linear and non-linear continuous covariates. We used a class of statistical models known as structured additive regression (STAR) models [27,28]. These models allowed us to estimate the effects of the different covariates on the observed response. In addition, the flexibility of the STAR models allowed us to simultaneously account for the spatial autocorrelation and heterogeneity, common with most geographically referenced data. This is called geo-additive models [29]. The model development is similar to that outlined in the appendix section of [25], however, for the purpose of clarity we give a description of the modelling approach below.

Let $y_{j}\in \left\{0,1\right\}$ denote the FGM/C status of girl j such that $y_{j}=1$ if girl j was cut and $y_{j}=0$ if girl j was uncut. In other words, $Y_{j}$ is a Bernoulli distributed random variable with realizations $y_{j}$ and with probability, $p_{j}$; $j=1,\ldots ,m_{i}$, where $m_{i}$ is the total number of girls in the geographical referenced region i. Thus, $y_{ij}\in \left\{0,1\right\}$ is the FGM/C status of girl j in region i; i=1,…,N, where N=14, the number of regions. Therefore, in all, the total number of girls in the entire population is given by $M=\underset{i}{\sum }m_{i}$. Then for our purposes, we give the logit link with function h(η)=μ=E[y|.], for the fully adjusted model based upon the pooled data as in (1) below:(1)$$\begin{array}{ll} \eta_{j}=logit\left(p_{j}\right) & =\log\left(\frac{p_{j}}{1-p_{j}}\right)\\ & =\beta_{0}+z_{j}^{\prime }\beta +f_{1}\left(x_{j1}\right)+\ldots +f_{p}\left(x_{jp}\right)+f_{t}\left(t\right)\;+\;f_{age}\left(age\right)\\ & +f_{pcut}\left(pcut\right)+f_{str}\left(s_{i}\right)+f_{unstr}\left(s_{i}\right)+f_{st}\left(s,t\right) \end{array}$$
where $f_{1}\left(.\right),\ldots ,f_{p}\left(.\right)$ are the functions (no necessarily smooth) of non-linear continuous covariates, $x_{j}s$ such as age, time effects etc. $\beta_{0}$ is the intercept, $\beta ={\left(\beta_{1},\;\ldots ,\;\beta_{p}\right)}^{\prime }$ are unknown coefficients of other class of covariates, $z_{j}s$; $s_{i}=1,\ldots ,S,$ where $s_{i}$ is the geographically referenced location of girl j, $f_{str}\left(.\right)$ and $f_{unstr}\left(.\right)$ denote the structured (correlated) and the unstructured(uncorrelated) spatial effects, respectively; $f_{t}\left(t\right),\;f_{st}\left(s,t\right),f_{age}\left(age\right),\;f_{pcut}\left(pcut\right)$ are smooth functions (assumed to be non-linear) of time (survey year), unobserved space-time interactions, age (mother or daughter), proportion of cut women in a given community, respectively.

For a full Bayesian inference using Markov Chain Mote Carlo (MCMC) techniques, we allowed for spatial autocorrelation and borrow strength from neighbouring regions by assigning Markov random fields prior [29]:(2)$$f_{str}\left(s\right)|f_{str}\left(r\right),\tau_{s};r\neq s∼N\left(\underset{r∼s}{\sum }\frac{f_{str}\left(r\right)}{N_{s}},\frac{\tau_{s}^{2}}{N_{s}}\right),\;$$
to the structured (or correlated) component of spatial effects, $f_{str}\left(.\right),\;\mathrm{where}\;N_{s}$ denotes the number of regions contiguous to region s; $\tau_{s}>0$ is a smoothness parameter, and r∼s implies that regions s and r are contiguous. To account for spatial heterogeneity, independent and identically distributed zero mean Gaussian prior was assigned to the unstructured (uncorrelated) spatial component, $f_{unstr}\left(.\right)$,
(3)$$f_{unstr}\left(s\right)|\tau_{u}∼N\left(0,\;\tau_{u}^{2}\right),\;$$
where $\tau_{u}>0$ is a variance parameter.

To the variance parameters, $\tau_{l}\;\left(l=s,u\right)$, we assigned inverse Gamma prior distribution that is, $\tau_{l}∼IG\left(a_{l},\;b_{l}\right)$, where $a_{.}$ and $b_{.}$ are hyperparameters.

To allow for the estimation of the smooth functions and the smoothing parameters, we model the various non-linear smooth functions using Bayesian P (enalised)-splines [30], a Bayesian analogue of the P-splines approach by Eilers and Marx [31]. The smooth function $f_{j}\left(x_{j}\right)$, is expressed as a linear combination of n B-spline basis functions as in (4):(4)$$f_{j}\left(x_{j}\right)\;=\;\underset{k}{\sum }\beta_{jk}B_{jk}\left(x_{j}\right).\;$$

The interaction smooth function $f_{st}\left(st\right)$ is modelled using the tensor product of one-dimensional B-splines
(5)$$f_{st}\left(st\right)=\underset{k}{\sum }\underset{j}{\sum }\beta_{jk\;}B_{sk}\left(s\right)B_{tj}\left(t\right),\;$$
for j,k=1, …,n, and we assign a first order random walk prior with a penalty matrix $K_{j}$ to the smoothness parameters as in (6) (see, [30]):(6)$$\beta_{j}|\tau_{j}^{2}\;\propto \exp\left(-\frac{1}{2\tau_{j}^{2}}\beta_{j}^{\prime }K_{j}\beta_{j}\right).\;$$

For implementation, we used BayesX (Georg-August-Universität Göttingen, Göttingen, Germany), an advanced statistical software, implemented in R statistical programming software through its R interface called R2BayesX [32]. Unknown parameters were estimated in a Markov Chain Monte Carlo (MCMC) technique [33]. For optimal performance of the MCMC, we set $a_{.}=1,\;b_{.}\;=0.0005$ and drew $T\left(=2\times 10^{4}\right)$ samples from the parameters space $\theta =\left\{\beta_{0},\;\left\{\beta_{j}\right\},\;\{f_{str}\left(.\right)\right\},\;\left\{f_{unstr}\left(.\right)\right\},\;\left\{p_{i}\right\}\}$ as well as the hyperparameters space $\phi =\left\{\tau_{\beta },\;\tau_{s},\;\tau_{u}\right\}$. For optimality, we used thinning and burn-in to improve the posterior estimates by storing only every 50th sampled value after discarding the first $B\left(=2\times 10^{3}\right)$ iterations as burn-in.

In the end, deviance information criterion (DIC) [34] was used for model selection and model fit assessment. For the DIC, the smaller the better. The posterior estimates obtained from the remaining T−B samples of the best fit models are reported in the appropriate tables, graphs and maps.

## 4. Results

### 4.1. Descriptive Analysis

Table 1 shows a total number of 43,155 girls aged 0–14 from 2005 SDHS (n = 11,878), 2010–2011 SDHS (n = 9740), 2015 SDHS (n = 7529) and 2017 SDHS (n = 14,008). This also indicates a national FGM/C prevalence of 20.4%, 11.9%, 14.6% and 14.0%, respectively, from 2005 to 2017. Other bivariate descriptive analyses results are shown in Table A1 of the Appendix A. The results showed that across the survey years the majority of the cut girls were aged between 5 to 14 years. In addition, before 2015, it was found that more daughters of older women were cut than the daughters of the younger mothers. The least prevalence of FGM/C was found among the daughters of women who never married and women who lived in the urban areas. Girls from Wolof and Serer ethnic groups as well as girls who lived in Diourbel, Fatick, Louga, Thios and Kaffrine regions, had the least prevalence of FGM/C across the years. In addition, FGM/C prevalence among girls whose mothers supported FGM/C continuation and whose mothers believed FGM/C was a religious dictate was at least 46% across the years.

### 4.2. Regional Evolution of FGM/C Prevalence among 0–14 Years Old Girls in Senegal

Figure 2 shows the evolution of FGM/C prevalence among Senegalese girls aged 0–14 years across the 14 administrative regions from 2005 to 2017. The highest prevalence of 78% was observed in Matam region followed by Kolda region (>65%) in 2005, while all the Central and Western regions including Dakar, Thies, Fatick, Diourbel and Kaolack had the least prevalence of not more than 17% in 2005. No values were recorded for Kedougou, Sedhiou and Kaffrine regions as they were only created after the 2005 SDHS. In 2010, prevalence in Matam and Kolda regions radically dropped to about 41% with highest prevalence of about 50% observed in the Casmance region of Sedhiou followed by the South Eastern region of Tambacounda (45%). In 2015, the observed prevalence in Matam over the previous five years increased to 57% followed by Sedhiou (55%) and Kolda (49%), while the Central and Western regions had the least prevalence. In 2017, observed prevalence in the Matam region remained high at around 61% while all the regions in the West and Centre remained very low in prevalence. Indeed, Figure 2 shows evidence of variations in the practice across the geographical locations of the respondents, thus, necessitating the need to unmask the key drivers of the observed spatial patterns.

### 4.3. Bayesian Hierarchical Geo-Additive Logistic Regression

Table 2 provides the brief descriptions of the models to the datasets along with the corresponding fit indices. Three different models were fitted to each of the datasets. To the most recent SDHS, 2017SDHS, we first fitted the unadjusted model or Model 1 which contains only the social norms ‘surrogate’ variables. These variables included the mother’s FGM/C status, the proportion of cut women, the woman’s support for the continuation of the practice, and the woman’s belief on whether FGM/C was a religious obligation. Secondly, we fitted the space-adjusted model or Model 2 which accounted for the unobserved effects of spatial locations and social norms simultaneously, with no recourse to other potential confounders such as religion and ethnicity. Thirdly, we fitted the so-called fully adjusted model or Model 3 to account for other potential confounders as well as unobserved spatial effects. The modelling strategy above was adopted for the pooled dataset within the coherent Bayesian hierarchical logistic regression model approach. Model 4, Model 5 and Model 6 were fitted to the pooled data. While Model 4 and Model 5 were equivalent to Model 1 and Model 2 fitted for the 2017SDHS, Model 6 included a temporal effect as well as space-time interaction in addition to total spatial effects and other potential confounders.

Preliminary analysis using Moran’s I test [35] showed evidence of spatial autocorrelation in the data. Results obtained from the models described above using the 2017 SDHS and the pooled 2010 to 2017 data are presented in Table 3 and Table 4, respectively.

### 4.4. 2017 Senegal Demographic and Health Surveys (SDHS)

Results in Table 3, show that across the three models fitted to the 2017SDHS, normative influences were found to be key FGM/C sustaining factors among 0–14 years old Senegalese girls. For example, the unadjusted model (Model 1) showed that a girl whose mother was cut was about 14.05 times (95% CI = 10.73, 18.30) more likely to be cut than her counterparts born to uncut mothers. After adjusting for other potential confounders (Model 3), the likelihood of cutting a girl whose mother was circumcised remained high with a posterior odds ratio, POR=14.74 (95% CI = 10.01, 21.31).

Figure 3 shows the non-linear effects of the community-level surrogate variables for social norms (proportion of cut women in the community) (Figure 3a) and the nonlinear effects of the ages of a girl (Figure 3b) and her mother (Figure 3c) obtained from Model 3 fitted on 2017 SDHS. We found that a girl’s likelihood of undergoing FGM/C increased as the number of cut women in her community increased and as her age increased. The highest likelihood of cutting was found among daughters of younger mothers (15–20 years) and daughters of older mothers (40–43 years).

Figure 4 shows the posterior maps of unobserved total spatial effects of a girl’s geographic location on her likelihood of experiencing FGM/C across the regions of Senegal. The top panel contains the spatial effects maps for Model 2 (mean (a) and 95% posterior probability (b)), while the bottom panel contains the spatial effect maps for the fully adjusted model (mean (c) and 95% posterior probability (d)). Based upon Model 2 (social norms variables and space), the posterior maps show that a girl who lived in Saint Louis, Matam, Tambacounda and Kolda regions, had significantly high risks of experiencing FGM/C, while a girl who lived in Dakar and Fatick regions had significantly low risk of being cut. However, after adjusting for other potential confounders including age, mother’s education, ethnicity and wealth index, the effect of living in Matam and Fatick became non-significant, while there existed significantly high risk of experiencing FGM/C by girls who lived in Saint Louis, Tambacounda and Kolda regions. This suggests that based upon the 2017 SDHS, social norms are key factors sustaining the practice of FGM/C in Saint Louis, Tambacounda and Kolda regions of Senegal.

Other factors considered in the fully adjusted model (Model 3) included some socio-demographic variables such as religion, marital status, ethnicity, mother’s occupation, mother’s highest level of educational attainment and type of place of residence (urban–rural). Results in Table 3 under Model 3 showed that a girl who lived in an urban region had a much lower likelihood of being cut than her rural area dwelling counterparts (POR = 0.52, 95% CI = 0.39, 0.68). A rather surprising result is that a girl who professed Muslim faith was less likely to be cut than her Christian counterparts; POR = 0.78 (95% CI = 0.38, 1.97). This is surprising in that most ethnic groups such as Mandigue and Poular (mainly Muslims) who practise FGM/C almost universally, did so on the notion that it was a religious dictate. However, we note that this finding highlights the strength of the approach we have adopted in this study in which we were able to account for the joint effects of the confounders coherently. It also signifies that based upon the 2017 SDHS, the effect of religion becomes not important in the presence of other confounders. Another interesting finding was that highest likelihood of being cut found among girls whose households were classified as belonging to the higher wealth index followed by middle wealth index household and least among girls in highest wealth index household, POR of 1.40 (95% CI = 0.93, 2.00) (for higher). In terms of ethnicity, the least likelihood of being cut was found among girls from Wolof and Serer ethnic groups, while girls who are from Diola ethnic group were 5.97 times (95% CI = 2.46, 16.59) more likely to be cut than their Wolof counterparts.

In addition, higher likelihood of being cut was found among girls whose mothers had up to secondary level of education than girls whose mothers attained up to higher level of education. Again, surprisingly, girls whose mothers attained, at most primary education, were less likely to experience FGM/C than their counterparts whose mothers had higher education. How can we explain this? This suggests that based upon evidence from the 2017 SDHS, the majority of the adherents of the practice were educated up to secondary level at least. We found lower likelihood of being cut among Senegalese girls whose mothers had formal employment than with girls whose mothers had informal or no jobs.

Based upon model fit indices as given in Table 2 above, Model 3 gave the best fit for the 2017 SDHS data. We, thus, give the fully adjusted predicted prevalence as well as the standard deviation for 2017 based upon Model 3 in Figure 5. The predicted fully adjusted mean prevalence map (b) shows highest fully adjusted FGM/C prevalence of 58.5%, 47%, 45% and 44% in Matam, Tambacounda, Sedhiou and Kedougou regions, respectively. However, we note a higher uncertainty in estimation for the Sedhiou and Kedougou regions (Figure 5c). Very low fully adjusted prevalence of less than 13% were predicted for the western and central regions of Dakar, Thies, Fatick, Kaffrine, Diourbel and Kaolack.

### 4.5. Pooled 2010 to 2017 SDHS Data

In Table 4, we show the results obtained from the Bayesian hierarchical spatial and spatio-temporal models fitted to the pooled 2010SDHS to 2017SDHS data. We found that across the models, surrogate variables of social norms indicated that social norms have been key drivers of FGM/C among 0–14 year old Senegalese girls over the years. For example, we found across the three models that daughters of cut women had a higher probability of being cut than their counterparts whose mother were uncut, with posterior odds ratio (POR) of at least 13.38. In addition, the likelihood of being cut among girls whose mothers supported FGM/C continuation was found to be constantly higher than those whose mothers never supported FGM/C continuation with POR values of at least 3.55 (95% CI = 3.24, 3.89) across the three models. Similarly, although much lower than the effects of a woman’s FGM/C status, we found that girls whose mothers believed FGM/C was a religious dictate had a higher likelihood of experiencing the practice than their counterparts.

Furthermore, based upon Model III (the fully adjusted space-time model with interaction), we found that a girl who lived in an urban area had a lower probability of being cut than her rural counterparts, POR = 0.63 (95% CI = 0.56, 0.73). We found no significant influence of religion on a girl’s likelihood of being cut. In terms of household wealth, we found that a girl whose household was classified as belonging to the higher wealth quintile had a higher likelihood of being cut than her middle wealth quintile household counterpart. This was similar to the finding under SDHS 2017. However, the least likelihood of being cut existed among girls from the highest wealth quintile household, but there is no significant difference between the likelihood of a girl who lived in the middle-class household and her counterparts in lower and lowest wealth quintile households.

In terms of ethnicity, again the least likelihood of being cut was found among Wolof and Serer girls, while girls from Soninke, Diola, Poular and Mandingue ethnic groups had a higher likelihood of being cut than their Wolof counterparts. Unlike the 2017 SDHS, results from the pooled data showed that girls whose mothers had no education (POR = 1.52, 95% CI = 0.95, 2.49) had higher likelihood of being cut than their counterparts whose mothers had higher education. However, no significant difference in likelihood existed between girls whose mothers had higher education and their counterparts whose mothers had primary and secondary education at most. Daughters of women who had a formal job had a lower likelihood of being cut than their counterparts whose mothers had informal or no jobs.

Figure 6 shows the posterior mean spatial effects maps with the corresponding 95% posterior probability maps obtained after adjusting for the unobserved effects of geographical locations only or Model 5 (top panel), and after adjusting for unobserved location effects and other potential confounders including temporal and space-time interactions Model 6 (bottom panel). The maps reveal important combined effects of the social norms factor with respect to geographical locations over the years. The posterior maps from Model 5 (top panel) show that a girl who lived in Saint Louis, Matam, Tambacounda and Kolda regions had significantly high risk of experiencing FGM/C while her counterparts who lived in Dakar, Thies and Fatick regions had a significantly low FGM/C risks.

After adjusting for the unobserved influence of time and space–time interactions as well as the influence of other potential confounders such as age, ethnicity, religion, wealth and education (Model 6), only a girl who lived in Matam region had a significantly high risk of being cut. This finding demonstrates the overall effects of the social norms factor with respect to geographical location over the years. It also illustrates the increased statistical power gained from the pooled data due to a decreased standard deviation thus revealing important variations that remained insignificant with one time point data.

In Figure 7, we show the graphical representations of the effects of the proportion of cut women in a girl’s community (Figure 7a), her age (Figure 7b) and her mother’s age (Figure 7c), on her likelihood of experiencing FGM/C. We found that a girl’s likelihood of experiencing FGM/C increased when the proportion of cut women in her community exceeded 30%, and also increased with her age. We also found that girls born to younger mothers aged 15–20 years had a higher likelihood of being cut than daughters of older mothers.

Based upon the fit indices in Table 2, Model 6 was identified as the best fit model for the pooled dataset. We present fully adjusted posterior prevalence maps for 2017 based upon Model III from the pooled data, as well as the standard deviation map in Figure 8. Again, we found the highest fully adjusted predicted posterior prevalence value of 60.4% in Matam region with smaller variance. The predicted posterior prevalence in Dakar, Thies, Kaolack, Fatick, Diourbel and Kaffrine remained low at less than 13%. Again, with the pooled data which allowed the incorporation of time and space-time interaction into the modelling, we have gained extra statistical power and improved parameter estimates.

Finally, in Figure 9, we show the fully adjusted temporal trend of FGM/C among 0–14 Senegalese girls obtained from Model 6 of the pooled data. Figure 8 shows that there was a radically higher likelihood of being cut in 2015 than in 2010. However, this was found to decrease significantly, albeit at a slower pace, in 2017.

## 5. Discussion

In this paper, we examined the roles of social norms in the persistence of FGM/C in Senegal despite years of concerted interventions. The key motivation is the need to test the theory which suggests that FGM/C is a social norm in that the shared norms between individuals or groups within a community, or between communities are potential factors sustaining the practice of FGM/C especially within sub-Saharan African countries [17,18,19]. To do this, we used a class of Bayesian hierarchical geo-additive regression models which allowed a simultaneous incorporation of geographical location covariates to capture differences in spatial autocorrelations, and other linear and non-linear covariates. We used a woman’s FGM/C status, her support for the continuation of the practice and her belief on the link between FGM/C and religion, as key individual-level surrogates for social norms. Besides, the proportion of cut women in a girl’s community served as the community-level surrogate variable. The models were fitted to the 2017 Senegal Demographic and Health Surveys (SDHS) data and a pooled dataset consisting of variables common to the 2010, 2015 and 2017 SDHS. We used BayesX software implemented in R statistical programming software through its R interface known as R2BayeX, to implement the models. Full Bayesian inference was carried out using MCMC techniques.

Our findings confirm the social norms hypothesis that social norms were key factors that sustained FGM/C in Senegal over the years. Evidence identified social norms as the key factors driving the persistence of FGM/C in Matam, Saint Louis, Tambacounda and Kolda regions. On the other hand, girls who lived in the regions of Dakar and Thies (West), and Fatick, Diourbel, Kaffrine and Kaolack (Centre), had the least likelihood of being cut. The ethnic groups with the highest probability of having their 0–14 year’s old girls cut included Soninke, Diola, Poular and Mandigue. The least likelihood of FGM/C was found among Wolof and Serer girls.

Results are also consistent with the concept of assimilative change and shifting reference groups [36], which involves processes of cultural borrowing or innovation, along with reassessment of prior norms. Assimilation change may occur more readily in communities with geographic proximity with non-practicing groups. For example, Thies, Fatick and Diourbel regions which are regions with the least prevalence and the lowest risk of FGM/C are mainly inhabited by Wolofs and Serers who are known to be largely FGM/C non-practising ethnic groups. The Poular people also appear to have been positively influenced by not cutting their girls probably because they live in the Louga region, which is geographically proximate to their counterparts in the Thies, Fatick and Diourbel regions. Their reference group members may include people from a variety of social circles: close friends and family, residents of one’s community, peers from school, work colleagues, and fellow members of a church or mosque. As confirmed by the social norm/convention theory, in order to change social norms, there is a need to change one’s reference group. Since different social norms may be influenced by different reference groups, such change should align with groups that tolerate the desired change. In other words, through social interactions between practising and non-practising families, people may have a greater opportunity to have different views about some age long traditions. A similar study on three Kenyan communities showed that higher inter-ethnic interaction was associated with a more dramatic shift away from norms in support of FGM/C [37]. In our study, the inhabitants of Kaffrine region, for example, are mainly the non-practising Wolof ethnic group, with the practising Poular ethnic group as the minority. It is most likely that the Poular ethnic group must have aligned with the desired change championed by their reference group (Wolof), thus jettisoning FGM/C in Kaffrine since there are no social sanctions for not conforming. In regions where there is negligible ethnic diversity, but with high risk of FGM/C, shifting to a non-practising reference group may be very difficult. However, where there is ethnic mixture involving people who do and do not practice FGM/C, such a shift may be possible, that is, a lower local prevalence of FGM/C may facilitate realigning one’s reference group little or no social sanctions. In addition, the finding that girls who lived in urban areas were less likely to be cut supports the hypothesis that norm change may be more likely to occur among people exposed to international norms opposing FGM/C, particularly through exposure to messages in the media [20,21].

The fully adjusted prevalence maps gave a clearer picture of the variations in the practice in Senegal with Matam region having the highest posterior prevalence in 2017, while Dakar, Fatick, Diourbel, Kaffrine and Kaolack had the least prevalence. A finding of lower risks of cutting a girl in western regions of Senegal may be explained by the influence of Tostan and many other organisations operating in those regions

Although, the advanced Bayesian hierarchical geo-additive regression models employed here allowed us to accurately capture the potential spatial (geographical) structure of the prevalence of FGM/C among 0–14 year old girls in Senegal, particularly in relation to the key individual- and community-level drivers of the practice. We note, however, that there exist two limitations of the study we identified. First, given that the information on a girl’s FGM/C status were provided by their mothers, the issues of inaccurate self-reporting largely due to difficulty in recalling an event that took place several years ago or due to the fear of being prosecuted by the state under the provisions of the anti-FGM/C legislation in Senegal. Secondly, we note that the datasets we analysed included information on girls aged 0–14 years only. Therefore, it is important to bear in mind while interpreting our results that the FGM/C status of a girl as at the time of the survey may not be her final status in that a girl who was not cut at age 14 may still be cut in the future. Nevertheless, it is highly unlikely that bias due to inaccurate reporting or recall bias will have a significant effect on the findings of this study. Besides, the statistical modelling approach employed in this study allowed the incorporation of multiple sources of variability in parameter estimation within a Bayesian hierarchical spatio-temporal regression modelling framework, thereby, allowing a straightforward quantification of uncertainties.

## 6. Conclusions

Social norm plays a significant role in the persistence of FGM/C in Senegal. We found that a woman’s FGM/C status, her support for the continuation of FGM/C and her belief that FGM/C is a religious obligation, were key determinants of her daughter’s FGM/C status. Specifically, daughters of cut women, daughters of women who would not want FGM/C to be stopped and daughters of women who believed that FGM/C was required by religion, were most likely to experience FGM/C at some point in their lives. There is a high likelihood of cutting a girl who lived in a community where more than 30% of the female inhabitants were circumcised. In such a situation, girls are pressured by their peers to conform and get cut or risk social exclusion. Results show that FGM/C in Senegal varied temporally and spatially (geographically), peaked in 2005 with Matam, Tambacounda and Kolda identified as FGM/C hotspot regions in Senegal, and also identified Poular, Soninke, Diola and Mandingue ethnic groups as the FGM/C high risk ethnic groups.

A tailored intervention should target the identified hotspot regions and ethnic groups in its design and implementation in partnership with religious, traditional and political leaders who are generally revered within their communities and thus most likely to champion changes that would potentially bring about a total abandonment of the practice in Senegal.

## Figures and Tables

**Figure 1 ijerph-18-03822-f001:**
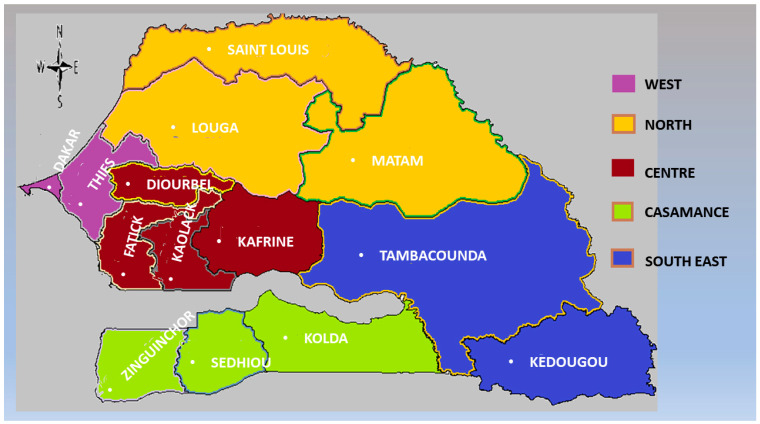
Map of Senegal showing the administrative regions (Source: authors).

**Figure 2 ijerph-18-03822-f002:**
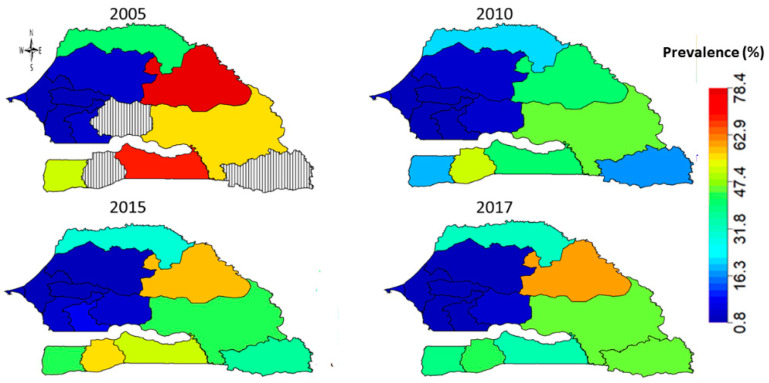
Evolution of FGM/C prevalence among girls aged 0–14 years across the 14 administrative regions of Senegal from 2005 to 2017.

**Figure 3 ijerph-18-03822-f003:**
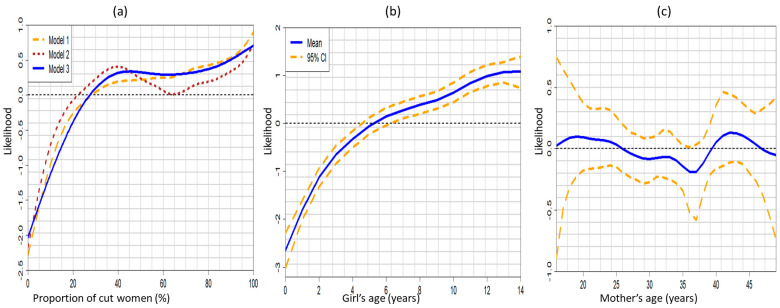
Non-linear effects of (**a**) the proportion of cut women in the community for the three models, (**b**) girl’s age, and (**c**) mother’s age, on a girl’s likelihood of experiencing FGM/C. Evidence from the 2017 SDHS (Model 3).

**Figure 4 ijerph-18-03822-f004:**
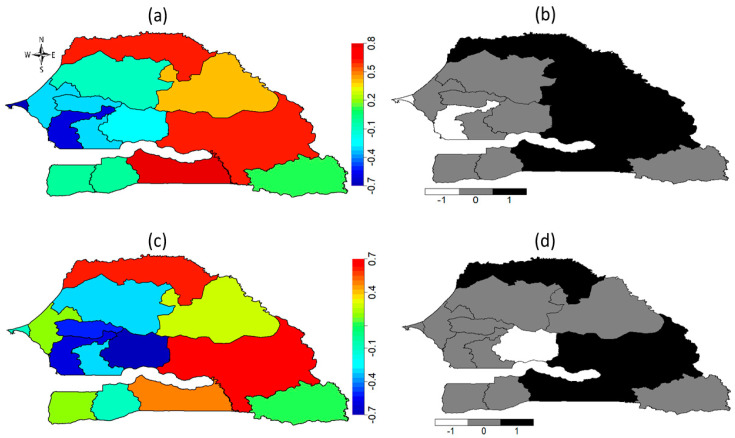
(**a**) Posterior risk map and (**b**) 95% posterior significance map based on Model 2; (**c**) Posterior risk map and (**d**) 95% posterior significance map based on Model 3. Deep blue to red corresponds to low risk to high risk. Black colour indicates significantly high-risk regions, white colour indicates significantly low risk regions and grey colour indicates non-significant regions. Evidence from 2017 SDHS.

**Figure 5 ijerph-18-03822-f005:**
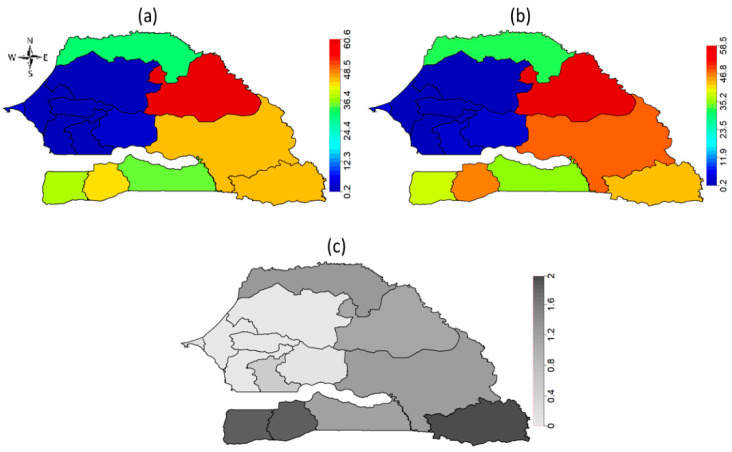
Observed prevalence (**a**), predicted posterior prevalence (**b**), and posterior standard deviation (**c**) of Senegalese 0–14 year old girls’ FGM/C. Deep blue to red corresponds to low to high prevalence. Light to dark grey colours correspond to increase in uncertainty of estimation. Evidence from 2017 SDHS (Model 3).

**Figure 6 ijerph-18-03822-f006:**
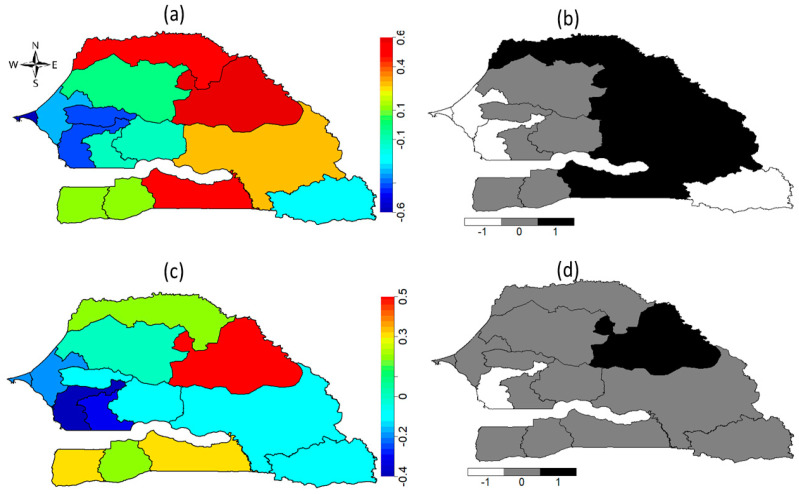
(**a**) Posterior risk map and (**b**) 95% posterior significance map based on Model 5; (**c**) Posterior risk map and (**d**) 95% posterior significance map based on Model 6. Deep blue to red corresponds to low risk to high risk. Black colour indicates significantly high-risk regions, white colour indicates significantly low risk regions and grey colour indicates non-significant regions. Evidence from the pooled 2010 to 2017 SDHS.

**Figure 7 ijerph-18-03822-f007:**
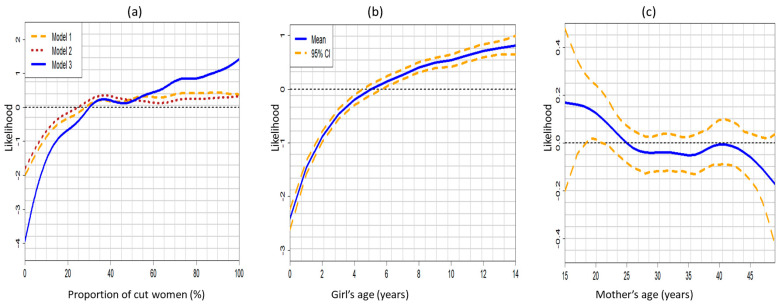
Non-linear effects of (**a**) the proportion of cut women in the community for the three models, (**b**) the girl’s age, and (**c**) the mother’s age, on a girl’s likelihood of experiencing FGM/C. Evidence from the pooled 2010 to 2017 SDHS data (Model 6).

**Figure 8 ijerph-18-03822-f008:**
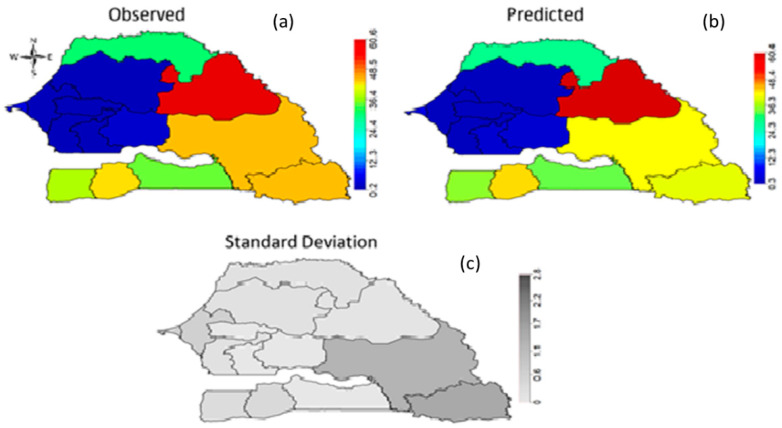
Observed prevalence (**a**), predicted posterior prevalence (**b**), and posterior standard deviation (**c**) of Senegalese 0–14 year old girls’ FGM/C. Deep blue to red corresponds to low to high prevalence. Light to dark grey colours correspond to increase in uncertainty of estimation. Evidence from the pooled 2010 to 2017 SDHS (Model 6).

**Figure 9 ijerph-18-03822-f009:**
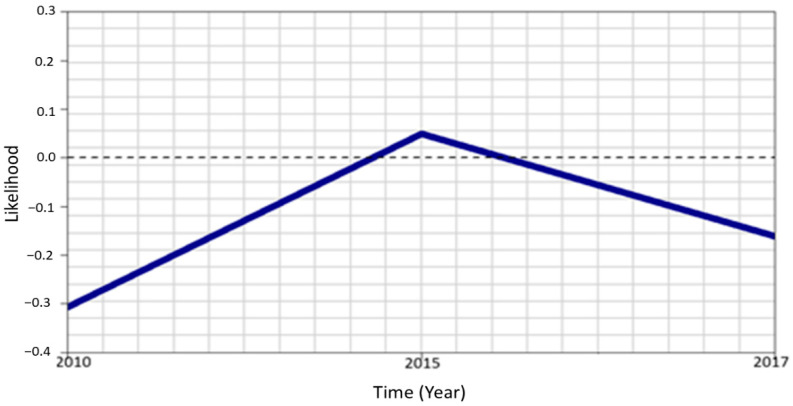
The trend of FGM/C over time. Evidence from the fully adjusted model fitted to combined data (Model 6).

**Table 1 ijerph-18-03822-t001:** Sample size of girls aged 0–14 years for each of the Senegal DHS surveys from 2005 to 2017 as well as the corresponding female genital mutilation/cutting (FGM/C) national prevalence.

Survey	Girls 0–14 Years	National Prevalence (%)
2005 SDHS	11,878	20.4
2010–2011 SDHS	9740	11.9
2015 SDHS	7529	14.6
2017 SDHS	14,008	14.0

Note: In the 2005 Senegal DHS, the FGM/C questions were asked about the most recently cut daughters of any age; for this analysis, sample size is limited to most recently cut girls aged 0–14. In the 2010–2011 Senegal DHS, the FGM/C questions were asked for all.

**Table 2 ijerph-18-03822-t002:** Deviance information criterion (DIC) from the three models fitted on the 2017 Senegal Demographic and Health Survey (SDHS) data and the combined 2010 to 2017 data.

Data	Model	Description	DIC
2017 SDHS	Model 1	Normative influence variables only	6427
	Model 2	Normative influence variables and total space	6329
	Model 3	Normative influence variables, space and other individual-level covariates	3542
2010 to 2017 pooled data	Model 4	Normative influence variables only.	15,300
	Model 5	Normative influence variables, space without time and space-time interaction.	15,160
	Model 6	Normative influence variables, space and time, space-time interaction and other individual-level covariates	12,324

**Table 3 ijerph-18-03822-t003:** Unadjusted and adjusted posterior odds ratios (POR) and associated 95% credible regions of circumcision of girls, ages 0–14, across selected covariates (Senegal, DHS 2017) from Bayesian geo-additive models.

Predictor	Level	Model 1 POR (95% CI)	Model 2 POR (95% CI)	Model 3 POR (95% CI)
DEMOGRAPHIC				
Place of residence	Rural (ref)			1.00
	Urban			0.52 (0.39, 0.68)
Religion				
	Christian (ref)			1.00
	Muslim			0.78 (0.33, 1.97)
Wealth index				
	Middle (ref)			1.00
	lower			0.89 (0.65, 1.20)
	lowest			0.94 (0.68, 1.31)
	Higher			1.40 (0.93, 2.00)
	Highest			0.81 (0.39, 1.63)
Ethnicity				
	Wolof (ref)			1.00
	Idiola			5.97 (2.46, 16.59)
	Mandingue			3.84 (1.88, 9.47)
	Non-Senegalese			2.15 (0.95, 5.16)
	Other			4.25 (1.86, 11.13)
	Poular			3.50 (1.76, 8.25)
	Serer			0.49 (0.10, 2.02)
	Soninke			3.74 (1.61, 10.47)
SOCIAL NORMS				
(SOCIAL NORMS) Mother cut				
	No (ref)	1.00	1.00	1.00
	Yes	14.05 (10.73,18.3)	14.19 (10.93, 19.01)	14.74 (10.01, 21.31)
Support continuation				
	Be stopped (ref)	1.00	1.00	1.00
	Continued	3.82 (3.32, 4.51)	3.88 (3.40, 4.49)	5.28 (4.36, 6.55)
	Depends	1.27 (0.90, 1.81)	1.29 (0.85, 1.80)	0.99 (0.55, 1.76)
FGM/C is required by religion				
	No (ref)	1.00	1.00	1.00
	Yes	1.69 (1.43, 1.96)	1.72 (1.48, 2.03)	1.94 (1.51, 2.39)
WOMEN’S AGENCY				
Mother’s education				
	Higher (ref)			1.00
	No education			0.77 (0.29, 2.46)
	Primary			0.87 (0.34, 2.70)
	Secondary			1.30 (0.47, 4.21)
WOMEN’S OPPORTUNITIES				
Mother’s occupation				
	Formal (ref)			1.00
	Informal			1.20 (0.93, 1.54)
	Not working			1.53 (0.97, 2.49)
Who decides?				
husband’s expenditure	Alone (ref)			1.00
	Husband has no earning			3.85 (1.12, 12.12)
	hus/partner/someone else			0.81 (0.36, 1.61)
	With hus/partner/someone else			1.29 (0.57, 2.88)
GENDER NORMS				
Female positive attitude to wife beating:				
Wife beating for going out				
	No (ref)			1.00
	Yes			0.90 (0.65, 1.26)
Wife beating for neglecting the children				
	No (ref)			1.00
	Yes			1.40 (0.96, 2.04)
Wife beating for arguing with the husband				
	No (ref)			1.00
	Yes			0.76 (0.49, 1.12)
Wife beating for denying husband sex				
	No (ref)			1.00
	Yes			1.54 (1.12, 2.12)
Wife beating for denying husband food				
	No (ref)			1.00
	Yes			0.74 (0.57, 0.96)
Who makes large household purchases	Alone (ref)			1.00
	Husband/partner			0.37 (0.17, 0.74)
	With husband/par			0.28 (0.13, 0.56)
Who makes decision on mother’s health	Alone(ref)			1.00
	Husband/partner			1.81 (1.00, 3.32)
	With husband/par			1.10 (0.58, 2.17)
MEDIA INFORMATION				
Read Newspaper	Not at all (ref)			1.00
	Less than once a week			0.36 (0.19, 0.67)
	At least once a week			0.57 (0.20, 1.53)
Listen to Radio	Not at all (ref)			1.00
	Less than once a week			1.18 (0.92, 1.49)
	At least once a week			1.19 (0.92, 1.60)
Watch Television	Not at all (ref)			1.00
	Less than once a week			0.76 (0.60, 0.96)
	At least once a week			0.76 (0.59, 1.01)

Model 1: Unadjusted Bayesian multivariate regression model. Model 2: Adjusted for unobserved total spatial effects. Model 3: Fully adjusted model. POR = posterior odds ratio; CI = credible interval.

**Table 4 ijerph-18-03822-t004:** Unadjusted and adjusted posterior odds ratios (POR) and associated 95% credible regions of circumcision of girls, ages 0–14, across selected covariates (pooled 2010 to 2017 SDHS) from Bayesian geo-additive models.

Predictor	Level	Model 4 POR (95% CI)	Model 5 POR (95% CI)	Model 6 POR (95% CI)
DEMOGRAPHIC				
Place of residence	Rural (ref)			1.00
	Urban			0.63 (0.56, 0.73)
Religion				
	Christian (ref)			1.00
	Muslim			0.95 (0.69, 1.33)
Wealth index				
	Middle (ref)			1.00
	lower			0.93(0.79, 1.12)
	lowest			0.91 (0.76, 1.11)
	Higher			1.17 (0.88, 1.52)
	Highest			0.84 (0.56, 1.20)
Ethnicity				
	Wolof (ref)			1.00
	Idiola			3.27 (1.95, 5.65)
	Mandingue			2.75 (1.74, 4.66)
	Non-Senegalese			2.35 (1.50, 4.03)
	Other			2.84 (1.83, 4.79)
	Poular			3.19 (2.08, 5.21)
	Serer			0.86 (0.42, 1.88)
	Soninke			4.24 (2.57, 7.70)
SOCIAL NORMS				
Mother cut				
	No (ref)	1.00	1.00	1.00
	Yes	13.69 (11.04, 17.13)	13.63 (11.19, 16.57)	13.38(10.56, 17.17)
Support continuation				
	Be stopped (ref)	1.00	1.00	1.00
	Continued	3.55 (3.24, 3.89)	3.59 (3.29, 3.95)	4.96 (4.43, 5.59)
	Depends	1.25 (0.99, 1.61)	1.22 (0.97, 1.56)	1.25 (0.91, 1.68)
FGM/C is required by religion				
	No (ref)	1.00	1.00	1.00
	Yes	1.39 (1.26, 1.54)	1.40 (1.27, 1.54)	1.64 (1.43, 1.89)
WOMEN’S AGENCY				
Mother’s education				
	Higher (ref)			1.00
	No education			0.85 (0.62, 1.19)
	Primary			0.78 (0.57, 1.07)
	Secondary			0.31 (0.05, 1.38)
Highest educational level of mother’s husband	Higher (ref)			1.00
	No education			1.52 (0.95, 2.49)
	Primary			1.05 (0.64, 1.70)
	Secondary			0.97 (0.57, 1.57)
WOMEN’S OPPORTUNITIES				
Mother’s occupation				
	Formal (ref)			1.00
	Informal			1.35 (1.17, 1.59)
	Not working			1.38 (1.13, 1.76)
Partner’s occupation	Formal (ref)			1.00
	Informal			0.98 (0.85, 1.11)
	Not working			0.69 (0.50, 0.97)
Woman currently employed				
	No (ref)			1.00
	Yes			0.75 (0.63, 0.90)
Who decides?				
husband’s expenditure	Alone (ref)			1.00
	Husband has no earning			1.69 (0.70, 4.84)
	hus/partner/someone else			0.66 (0.45, 0.96)
	With hus/partner/someone else			0.76 (0.48, 1.16)
GENDER NORMS				
Female positive attitude to wife beating:				
Wife beating for going out				
	No (ref)			1.00
	Yes			1.08 (0.90, 1.28)
Wife beating for neglecting the children				
	No (ref)			1.00
	Yes			1.02 (0.86, 1.24)
Wife beating for arguing with the husband				
	No (ref)			1.00
	Yes			0.95 (0.79, 1.14)
Wife beating for denying husband sex				
	No (ref)			1.00
	Yes			1.38 (1.17, 1.62)
Wife beating for denying husband food				
	No (ref)			1.00
	Yes			0.92 (0.80, 1.05)
Who makes large household purchases	Alone (ref)			1.00
	Husband/partner			0.96 (0.67, 1.36)
	With husband/partner			0.87 (0.62, 1.27)
Who makes decision on mother’s health	Alone(ref)			1.00
	Husband/partner			1.25 (0.92, 1.68)
	With husband/partner			0.96 (0.67, 1.30)
MEDIA INFORMATION				
Read Newspaper	Not at all (ref)			1.00
	Less than once a week			0.74 (0.52, 1.05)
	At least once a week			0.96 (0.56, 1.67)
Listen to Radio	Not at all (ref)			1.00
	Less than once a week			1.31 (1.13, 1.53)
	At least once a week			1.37 (1.19, 1.57)
Watch Television	Not at all (ref)			1.00
	Less than once a week			1.01 (0.80, 1.14)
	At least once a week			0.96 (0.80, 1.14)

Model 4: Unadjusted Bayesian multivariate regression model. Model 5: Adjusted for unobserved total spatial effects. Model 6: Fully adjusted model. POR = posterior odds ratio; CI = credible interval.

## Data Availability

Data used in this study are freely available on the DHS website (https://dhsprogram.com/data/ accessed on 13 February 2021) and is easy to download with permission from the DHS.

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
