# Peer review of "Analysing Normative Influences on the Prevalence of Female Genital Mutilation/Cutting among 0–14 Years Old Girls in Senegal: A Spatial Bayesian Hierarchical Regression Approach"

_ijerph, 2021, doi:10.3390/ijerph18073822_

Round 1

Reviewer 1 Report

Dear authors,

Thank you very much for your manuscript about FGM/C in Senegal. It was really an (disturbingly) enjoyable read. I have no large comments or complaints - though as I informed the editor, I am not well versed in Bayes, and so I leave any comments re: Bayes to hopefully other reviewers. 

My only comment is small. On page 3 lines 130-131, you state "We note that this Bayesian hierarchical modelling approach popular in disease mapping studies [21, 22] has only been sparingly applied in FGM/C related studies [15, 22-25]." First, I think you should probably add a comma - approach, popular in disease mapping studies, has... But more importantly, the sentence does not fit with the citations - you state it is popular, but provide 2 sources, while saying sparingly, and provide 4. The reader would interpret that it seems more popular in FGM/C and sparingly in disease mapping.  This is a very well written paper. Another typo is under competing interests ("articl" should be "article."

Author Response

Reviewer 1’s comments:

Dear authors,

  1. Thank you very much for your manuscript about FGM/C in Senegal. It was really an (disturbingly) enjoyable read.

Response: Thank you.

I have no large comments or complaints - though as I informed the editor, I am not well versed in Bayes, and so I leave any comments re: Bayes to hopefully other reviewers. 

  1. My only comment is small. On page 3 lines 130-131, you state "We note that this Bayesian hierarchical modelling approach popular in disease mapping studies [21, 22] has only been sparingly applied in FGM/C related studies [15, 22-25]." First, I think you should probably add a comma - approach, popular in disease mapping studies, has

Response: Thank you. We have inserted a ‘comma’ after ‘approach’ in line 130 of the revised manuscript.

  1. ... But more importantly, the sentence does not fit with the citations - you state it is popular, but provide 2 sources, while saying sparingly, and provide 4. The reader would interpret that it seems more popular in FGM/C and sparingly in disease mapping. This is a very well written paper.

Response: Thank you. We have reworded the affected sentence in line 131 of the main manuscript to “has also been applied recently in FGM/C related studies”.

  1. Another typo is under competing interests ("articl" should be "article."

Response: Thank you. We have added the omitted ‘e’ to the affected word in line 508 of the main manuscript. 

Reviewer 2 Report

The authors report the results of a sophisticated statistical study of regional varitions in FGM/C within Senegal. They find significant variations among regions. They found that a girl’s probability of cutting was higher if her mother was cut, supported FGM/C continuation or believed that the practice was a religious obligation. In addition, living in rural areas and being born to a mother from certain ethnic groups increased a girl’s likelihood of being cut. The authors claim that their results should  prove useful in informing evidence-based intervention policies designed to achieve the abandonment of the practice in Senegal.

The introduction to the paper provides sufficient background. I am not an expert on Senegal or FGM/C. However, I was able to follow the introduction and found the references to the literature to be numerous, relevant, and up-to-date. I assume that it is reasonably comprehensive as well. I know that I earned a lot from reading it. I did not know, for example, that Senegal’s penal code criminalized FGM in 1999. The authors also describe several other ongoing policy interventions aimed at eradicating FGM. I would like to know whether the law on the books is actually implemented. Have there been prosecutions, convictions, and penalties imposed? There are accounts in the literature about other practices that violate human rights where the gap between formal laws and actual government practices is large. If the authors are aware of any previous research providing evidence that the FGM eradication policies in Senegal have been sincere it would be useful to cite that evidence.

The author’s use of social norms theory is interesting and directly relevant to the research question. 

The research design is appropriate and the statistical analysis is cutting edge. The methodology is adequately described. The data are available for others to use. Replication is possible. The authors  used individual and community-level characteristics along with the variable that identifies the geographical location of a girl and her mother. They also used mother’s location (urban vs rural, region of residence) to assess the influence of social interaction between two neighbouring regions. Variables used as proxies for social norms included individual-level (a woman’s FGM/C status, a woman’s support for FGM/C continuation, a woman’s belief about FGM/C being a religious dictate) and community-level (proportion of cut women in the community) characteristics. Other exposure variables included mother’s age, level of education, household wealth and religion.

 The results are clearly presented. Among the most interesting and surprising results was that FGM/C prevalence among girls whose mothers supported FGM/C continuation and whose mothers believed FGM/C was a religious dictate was at least 46% across the years. 

The conclusions are supported by the results. They support the social norms hypothesis. I was surprised by the hesitance of the authors to generalize beyond Senegal. I think several of their generalizations would hold in other settings. They were too modest. 

Overall, this is an excellent article. Once published, I will cite it in my own future papers. The only suggestion I have is that the authors better explain the policy implications of their findings. The abstract claims that their findings help inform the making of policy interventions that would more effectively eradicate FGM in Senegal, but they are not specific about what form those interventions should take. 

Author Response

Reviewer 2’s comments:

  1. The authors report the results of a sophisticated statistical study of regional varitions in FGM/C within Senegal. They find significant variations among regions. They found that a girl’s probability of cutting was higher if her mother was cut, supported FGM/C continuation or believed that the practice was a religious obligation. In addition, living in rural areas and being born to a mother from certain ethnic groups increased a girl’s likelihood of being cut. The authors claim that their results should prove useful in informing evidence-based intervention policies designed to achieve the abandonment of the practice in Senegal.

Response: Thank you. Yes, it is our hope that the outcomes of this research would provide policy managers the required statistical evidence for the development and implementation of more focused and tailored interventions, aimed at achieving an accelerated abandonment of FGM/C in Senegal.

  1. The introduction to the paper provides sufficient background. I am not an expert on Senegal or FGM/C. However, I was able to follow the introduction and found the references to the literature to be numerous, relevant, and up to date. I assume that it is reasonably comprehensive as well. I know that I earned a lot from reading it. I did not know, for example, that Senegal’s penal code criminalized FGM in 1999. The authors also describe several other ongoing policy interventions aimed at eradicating FGM. I would like to know whether the law on the books is actually implemented. Have there been prosecutions, convictions, and penalties imposed? There are accounts in the literature about other practices that violate human rights where the gap between formal laws and actual government practices is large. If the authors are aware of any previous research providing evidence that the FGM eradication policies in Senegal have been sincere it would be useful to cite that evidence.

Response: Thank you. Yes indeed, citing such potentially interesting works would be useful. However, as the focus of the present manuscript is primarily on the use of a novel approach to generating statistical evidence of the prevalence of the practice, we believe that the references therein would suffice. A recent scoping review study by the authors which also highlights the current state of the awareness and implementation of legislations against the practice of FGM/C across various jurisdictions of certain sub-Saharan African countries including Senegal (under review), has a rich pool of such potentially interesting references.

  1. The author’s use of social norms theory is interesting and directly relevant to the research question.

Response: Thank you.

  1. The research design is appropriate and the statistical analysis is cutting edge. The methodology is adequately described. The data are available for others to use. Replication is possible. The authors used individual and community-level characteristics along with the variable that identifies the geographical location of a girl and her mother. They also used mother’s location (urban vs rural, region of residence) to assess the influence of social interaction between two neighbouring regions. Variables used as proxies for social norms included individual-level (a woman’s FGM/C status, a woman’s support for FGM/C continuation, a woman’s belief about FGM/C being a religious dictate) and communitylevel (proportion of cut women in the community) characteristics. Other exposure variables included mother’s age, level of education, household wealth and religion.

Response: Thank you.

  1. The results are clearly presented. Among the most interesting and surprising results was that FGM/C prevalence among girls whose mothers supported FGM/C continuation and whose mothers believed FGM/C was a religious dictate was at least 46% across the years. 

Response: Thank you.

  1. The conclusions are supported by the results. They support the social norms hypothesis. I was surprised by the hesitance of the authors to generalize beyond Senegal. I think several of their generalizations would hold in other settings. They were too modest.

Response: Thank you. Indeed, the findings from the study could be generalised within FGM/C contexts across similar geographical and socio-economic settings. However, extrapolation to other contexts which are geographically, culturally, and socio-economically dissimilar (with wide variations) should be made with caution having in mind that the advanced statistical models (Bayesian hierarchical geo-additive models) employed in the analysis, incorporated geographical (spatial) structures as well as socio-cultural and socio-economic variables within the modelling framework.   

  1. Overall, this is an excellent article. Once published, I will cite it in my own future papers. The only suggestion I have is that the authors better explain the policy implications of their findings. The abstract claims that their findings help inform the making of policy interventions that would more effectively eradicate FGM in Senegal, but they are not specific about what form those interventions should take.

Response: Thank you. The Abstract contains only the highlights of the manuscript reflecting through the problem statement down to the discussion within the maximum allowable word count, thus, limiting the amount of information to be included therein. However, clear suggestions as to how to approach future interventions aimed at eliminating FGM/C in Kenya were provided in the Conclusion Section of the manuscript. Specifially, in lines 584 – 586, the authors wrote “These identified regions and FGM/C –practising ethnic groups should be targeted in the design and implementation of elimination intervention programmes. Such interventions must involve religious, traditional and political leaders.

Reviewer 3 Report

This is presented clearly for the most part, although there are a number of typos that should be fixed before publication (for example, it should be "still practiced" in line 56 and no "an" in "an evidence" in line 121; I'm not sure the question mark is appropriate at the end of line 98; "exclusions" need not be plural in line 579). 

Aside from grammatical issues, my only other concern has to do with the inadequate attention to gender roles and the prevalence of sexism/patriarchy/misogyny in shaping the social norms around FGM/C. What the authors present seems fairly obvious: FGM/C is more common in the daughters of cut women or where it has become the social norm -- out of fear of exclusion. A more compelling presentation would highlight the role of gender (in society and in religion, including Islam, since that seems to be the dominant religion) and its role in shaping views about this practice. The questions about agreeing with wife-beating could/should be used to explore not just correlation, but if perhaps causation is in play. The causes, effects, and possible solutions to prevent violence against women and girls should be the real focus; otherwise this seems like minutiae. 

This seems like a missed opportunity to study the beliefs and values informing these practices (as normative), which could be illuminating for better understanding the kinds of goals and strategies that could make progress toward these SDGs. 

Author Response

  1. This is presented clearly for the most part, although there are a number of typos that should be fixed before publication (for example, it should be "still practiced" in line 56 and no "an" in "an evidence" in line 121; I'm not sure the question mark is appropriate at the end of line 98; "exclusions" need not be plural in line 579).

Response: Thank you. We have included the omitted letter ‘d’ in ‘practice’ as found in line 56 of the main manuscript. The article ‘an’ has been removed from ‘an evidence’ in line 121. The question mark at the end of the sentence in line 98 should not be there and has now been removed, and ‘exclusions’ in line 579 is now written as ‘exclusion’.

  1. Aside from grammatical issues, my only other concern has to do with the inadequate attention to gender roles and the prevalence of sexism/patriarchy/misogyny in shaping the social norms around FGM/C. What the authors present seems fairly obvious: FGM/C is more common in the daughters of cut women or where it has become the social norm -- out of fear of exclusion.

Response: Thank you. At individual-level, a woman’s FGM/C status, her support for the continuation of the practice and her belief on the link between FGM/C and religion, were used as the key individual-level surrogates for social norms within the Bayesian hierarchical regression modelling framework. While at the community-level, the proportion of cut women in a girl’s community served as the social norm factor surrogate variable. Indeed, the rexults from the study revealed that FGM/C is still being sustained in Senegal largely due to social norm. According to Mackie et al.[1] FGM/C is a social norm which spread and became locked in place by interdependent expectations regarding marriageability.  This suggests that most women circumcise in order to meet the desires of men and become marriageable as circumcised women are deemed to be more likely to uphold marital fidelity. For this reason, FGM/C is seen as a prerequisite to marriage such that women who are uncut are deemed not marriageable and often pay the high price of foregoing marriage, legitimate childbearing or even become ostracized. FGM/C is, therefore, seen as a requirement not only for enhancing women’s chances of marriage but also for being socially acceptable within the social networks of cut women. One can argue that this notion emanated from the internalized influence of sexism/patriarchy/misogyny within the society, but a further research dedicated to the roles of gender and peer influence in perpetuating the practice would be interesting.  Here, we relied on the information/variables provided in the rich datasets from Senegal Demographic and Health Surveys (SDHS) covering from 2005 to 2017.

  1. A more compelling presentation would highlight the role of gender (in society and in religion, including Islam, since that seems to be the dominant religion) and its role in shaping views about this practice. The questions about agreeing with wife beating could/should be used to explore not just correlation, but if perhaps causation is in play. The causes, effects, and possible solutions to prevent violence against women and girls should be the real focus; otherwise this seems like minutiae.

Response: Thank you.

Although the present study found no statistically significant effects of wifebeating on a daughter’s likelihood of experiencing FGM/C at least for the datasets we analysed (see the ‘Gender Norms’ Section of Table 3A of the main manuscript), it might worth having a more focused research to further explore the roles of gender norms on the prevalence of FGM/C.

  1. This seems like a missed opportunity to study the beliefs and values informing these practices (as normative), which could be illuminating for better understanding the kinds of goals and strategies that could make progress toward these SDGs.

Response: Thank you. However, the authors strongly believe that the statistical evidence generated in this study have shown the progress made in the course of achieving FGM/C abandonment in Senegal in line with the SDGs, especially Target 5.3, which aims to eliminate all forms of violence against women and girls by the year 2030. The study also highlighted the potential challenges on the abandonment efforts and went a step further to suggest effective ways to consider while designing and implementing any future interventions. 

[1] Mackie, G.; LeJeune, J. Social Dynamics of Abandonment of Harmful Practices (2009).  A New Look at the Theory; Innocenti Working Paper No. 2009-06, Florence, UNICEF Innocenti Research Centre.

Reviewer 4 Report

The authors make use in the article of spatial Bayesian hierarchical models in order to analyze the normative influences on the prevalence of female genital mutilation/cutting among 0-14 year-old girls in Senegal, employing Markov chain Monte Carlo (MCMC) techniques for a full Bayesian inference, and assessing the models fit and complexity by means of the deviance information criterion (DIC).

The article includes a detailed introduction on the topic of female genital mutilation/cutting in Senegal, a section on the materials and methods used in the research, a section describing the statistical analysis used in the approach, a section with the results obtained, plus discussion and conclusion sections. The document is well written and with only a few language incorrections that do not affect its interpretation, and some details and aspect that might require additional explanation. These issues are pointed at in the following comments and suggestions.

1. It is somehow confusing that the texts are indented while the figures and tables are not so.

2. Some word spacing issues are present in the document, for example in lines 199, 220, 234, 235, 245, 269, 318, 325, '1OR' in header of Table 2A, 'AOR', 'BOR' and 'COR' in header of Table 3A, 341, 344, 419, 461, 464, 508, 565, 584, 592.

3. The intention of the use of bold type in the document is not clear, see lines 229, 232, 233, 267, 268, 343.

4. Line 128: The use of Bayesian hierarchical models is not properly justified, and it is not explained if other alternative approaches were considered and why they were rejected.

5. Line 196: Maybe it could be explained how the variables used as proxies for social norms or 'exposure variables' were chosen: from which set of variables and using which criteria. Also, some reference literature would be interesting to have.

6. Line 230: The equation seems to be broken in two because of the line spacing.

7. Line 265: Maybe 'techniques' should be written 'technique'.

8. Line 268: 'Burn-in' should not be capitalized.

9. Line 272: It seems that the sentence 'For the DIC' seems to be incomplete, or to be redundant.

10. Line 277: Perhaps 'shows total' should be written as 'shows a total', and commas used to separate thousands in the figures.

11. Line 278: Maybe 'indicates national' could be written as 'indicates a national', and commas used to separate thousands in the figures.

12. Line 279: In '14.%' a decimal figure seems to be missing, or the decimal point is redundant.

13. Line 291: In the Table A1, a number of comparative figures such as '<0.001' do not have a label associated to it. Also, commas should be used to separate thousands in the figures.

14. Lines 297 and 310: There are references to a 'Figure 2' that cannot be found in the document. Also, in the last line the text 'Indeed Figure 2, shows' should be written as 'Indeed, Figure 2 shows'.

15. Lines 312 to 327: References are made to 'Model 1', 'Model I', 'Model 2', 'Model II', 'Model 3' and 'Model III' to refer, it seems, to only three models. The use of Roman numeral happens also, for example, in pages 20, 21 y 22.

16. Table 2A: The reference to 'See graph' is not informative enough. Also, the heigth of rows seems to change without a particular reason. Additionally, at the bottom of page 12 the 'hus/partner/someone else' reference seems to be repeated at the top of page 13, with different values associated to it.

17. Line 337: The word 'Girls' should not be capitalized.

16. Table 2A: The reference to 'See graph' in page 14 is not informative enough. Also, the heigth of rows seems to change without a particular reason. Additionally, near the bottom of page 15 the 'hus/partner/someone else' reference seems to be repeated, with different values associated to it.

17. The title of sub-section 4.4 should be reviewed.

18. In Figures 3 and 7, the legend is missing in the graphs on the right.

19. Figure 5 seems to be displaced to the right and not fully visible.

20. Lines 422, 429, 435 and 444: Perhaps 'had higher' should be written as 'had a higher'.

21. Line 426: Maybe 'continuations with POR of' could be written as 'continuation with POR values of'.

22. Line 433: Perhaps 'terms household' could be written as 'terms of household'.

23. Line 437: Maybe 'from Highest' should be written as 'from the Highest'.

24. Line 439: Perhaps 'lower and lowest' could be written as 'Lower and Lowest'.

25. Line 446: Maybe 'among girls' should be written as 'between girls'.

26. Lines 447-448: Perhaps 'had formal job had lower' should be written as 'had a formal job had a lower', and 'had informal or' as 'had an informal one, or'.

27. Lines 454-455 and 468-469: Maybe 'of social norms factor with respect to geographical' could be written as 'of the social norms factor with respect to the geographical'.

28. Line 456: Maybe 'lived Saint' could be written as 'lived in Saint'.

29. Lines 457, 458 and 467: Perhaps 'had significantly' could be written as 'had a significantly'.

30. Line 470: Maybe 'to decreased' should be written as 'to a decreased'.

31. Line 473: Perhaps 'of proportion' could be written as 'of the proportion'.

32. Line 477: Maybe 'had highest' should be written as 'had a higher'.

33. Line 482: Please correct 'Tabe 2' to 'Table 2'.

34. Line 485: Perhaps 'prevalence of' could be written as 'prevalence value of'.

35. In Figure 8, the (a), (b) and (c) labels are missing.

36. Figure 9 seems to be oversized.

37. Line 528: Please correct 'counterpatcs' to 'counterparts'.

38. Line 551: Maybe 'gave clearer' could be written as 'gave a clearer'.

39. Line 555: Perhaps 'that regions' could be written as 'those regions.'.

40. Lines 556-567: The reference to the weakness in the study derived from it being reliant on the accuracy of the self-reportation of mothers, maybe affected by fear to prosecution, is very understandable and might compromise the relevance of its findings. Perhaps the results could be matched to those from other studies that would allow cross-checking in order to support the claims made, or else to include more objective complementary variables -perhaps coming from SHS as well- into the study to increase its strength.

41. Line 578: Perhaps 'pressurized' could be written to 'pressured'.

42. Line 579: Please correct 'exclusions.' to 'exclusion.'.

43. Line 583: Maybe 'hotspots regions' could be written as 'hotspot regions'.

44. Lines 585-586: The claim that 'Such interventions must involve religious, traditional and political leaders' maybe would require further elaboration to explain its intention and justification.

45. Line 593: The reference to 'to reduce the risk in FGM/C outcomes.' is not clear, and perhaot could be rephrased.

46. Line 598: Perhaps 'by Population' could be written as 'by the Population'.

47. Line 608: This line seems to be incomplete.

48. REFERENCES: It is recommended to check the authors' names abbreviations, as they are shown using different formats on different references.

Author Response

Reviewer 4’s comments:

The authors make use in the article of spatial Bayesian hierarchical models in order to analyze the normative influences on the prevalence of female genital mutilation/cutting among 0-14 year-old girls in Senegal, employing Markov chain Monte Carlo (MCMC) techniques for a full Bayesian inference, and assessing the models fit and complexity by means of the deviance information criterion (DIC).

The article includes a detailed introduction on the topic of female genital mutilation/cutting in Senegal, a section on the materials and methods used in the research, a section describing the statistical analysis used in the approach, a section with the results obtained, plus discussion and conclusion sections. The document is well written and with only a few language incorrections that do not affect its interpretation, and some details and aspect that might require additional explanation.

Response: Thank you.

These issues are pointed at in the following comments and suggestions.

  1. It is somehow confusing that the texts are indented while the figures and tables are not so.

Response: Thank you. The structure of the manuscript follows a submission template and the final published (if accepted) product will always follow the journal specifications. 

  1. Some word spacing issues are present in the document, for example in lines 199, 220, 234, 235, 245, 269, 318, 325, '1OR' in header of Table 2A, 'AOR', 'BOR' and 'COR' in header of Table 3A, 341, 344, 419, 461, 464, 508, 565, 584, 592.

Response: Thank you. The affected paragraphs and lines (199, 220, 234, 245, 269, 318, & 325) have been reformatted and the spaces removed. As a result, all other space issues have been resolved. We note the typographical error in Table 2A where AOR, BOR & COR are typographical versions of the posterior odds ratio (POR). This is line corrected for both Tables 2A and 3A.

  1. The intention of the use of bold type in the document is not clear, see lines 229, 232, 233, 267, 268, 343.

Response: Thank you. The bold forms of the letters are mathematical modelling ways of specifying a vector, that is, a variable with potentially multiple values why the individual values are not bolded as was the case of  and  as used in the manuscript.

  1. Line 128: The use of Bayesian hierarchical models is not properly justified, and it is not explained if other alternative approaches were considered and why they were rejected. Response: Thank you. The reason behind the choice of the advanced Bayesian hierarchical regression model employed in the study were highlighted in lines 124 – 136. This includes the need to account for both spatial autocorrelation and adjust for key covariates at the same time. It also enabled a straightforward approach of quantifying uncertainties in parameters estimation. This was further justified in lines 207 – 212 of the manuscript. Prominent among the reasons is the fact that the 2-stage cluster sampling technique utilised in the Senegal Demographis and Health Surveys (SDHS) does not guarantee independence of observational units. As a result, any statistical modelling approach that assumes independence of observations while ignoring the potential spatial autocorrelated is no longer appropriate. The Bayesian hierarchical geo-additive regression used here incorporated the spatial structures of the respondent’s place of residence into the model whilst simultaneously adjusted for other key covariates.
  2. Line 196: Maybe it could be explained how the variables used as proxies for social norms or 'exposure variables' were chosen: from which set of variables and using which criteria. Also, some reference literature would be interesting to have.

Response: The social norm ‘surrogate’ variables (a woman’s FGM/C status, her belief that FGM/C is a religious requirement, and the proportion of cut women within her community) were selected from the data based on a set of theories (e.g., Mackie, 1996; Mackie & LeJeun, 2009[1]; Shell-Duncan et all, 2011[2]) and have been applied in the context of Senegal (e.g., Matanda et al[3].). The social theories hypothesised that FGM/C is being sustained as a result of social norms which raise the expectations that women be circumcised in order to be marriageable and ‘clean’, or they will risk being ostracized by other circumcised women. As a result, women who do not wish to be excluded from the social networks of the other cut women who are generally powerful and very influential become obliged to comply. Therefore, it is not out of place to assume that a woman’s FGM/C status and other beliefs around FGM/C as well as the magnitude of direct or indirect pressure to conform from among other cut women within her community are among the closest potential factors perpetuating FGM/C as a social norm. These factors were found to be significant in our analysis. Also all the references mentioned herein are appropriately referenced in the main manuscript (Refs- 17 - 20)..

  1. Line 230: The equation seems to be broken in two because of the line spacing.

Response: Thank you. Equation 1 now reformatted to align the equations properly (see lines 228 - 230).

  1. Line 265: Maybe 'techniques' should be written 'technique'.

Response: Thank you. Changed to ‘technique’ on line 265.

  1. Line 268: 'Burn-in' should not be capitalized.

Response: Thank you. Changed to ‘burnin’ in line 268.

  1. Line 272: It seems that the sentence 'For the DIC' seems to be incomplete, or to be redundant.

Response: Thank you. The omitted statement “the smaller the better” is now inserted in line 272.

  1. Line 277: Perhaps 'shows total' should be written as 'shows a total', and commas used to separate thousands in the figures.

Response: Thank you. Changed to ‘show a total’ in line 277.

  1. Line 278: Maybe 'indicates national' could be written as 'indicates a national', and commas used to separate thousands in the figures.

Response: Thank you. Now included an ‘a’ in between ‘indicates national’.

  1. Line 279: In '14.%' a decimal figure seems to be missing, or the decimal point is redundant.

Response: Thank you. The missing 0 now added.

  1. Line 291: In the Table A1, a number of comparative figures such as '<0.001' do not have a label associated to it. Also, commas should be used to separate thousands in the figures.

Response: Thank you. ‘p’ which represents p-value now included wherever necessary throughout Table A1.

  1. Lines 297 and 310: There are references to a 'Figure 2' that cannot be found in the document. Also, in the last line the text 'Indeed Figure 2, shows' should be written as 'Indeed, Figure 2 shows.

Response: Thank you. The omitted Figure 2 is now inserted, and a comma has also been added in lines 312 to 313 of the main manuscript. The comma placed after Figure 2 has now been removed in line 310.

  1. Lines 312 to 327: References are made to 'Model 1', 'Model I', 'Model 2', 'Model II', 'Model 3' and 'Model III' to refer, it seems, to only three models. The use of Roman numeral happens also, for example, in pages 20, 21 y 22.

Response: Thank you. For ease of exposition, we used the two systems of naming to differentiate between the models fitted to the two datasets, namely, the single year dataset (2017) and the pooled dataset. Models I to III were fitted to the pooled datasets while Models 1 to 3 were fitted to the 2017 dataset. 

  1. Table 2A: The reference to 'See graph' is not informative enough. Also, the heigth of rows seems to change without a particular reason. Additionally, at the bottom of page 12 the 'hus/partner/someone else' reference seems to be repeated at the top of page 13, with different values associated to it.

Response: Thank you. We have replaced ‘see graph’ with ‘see Figure 3b’ and ‘see Figure 7b’ at the affected points. The factor level 'With hus/partner/someone else' has now replaced the repeated 'hus/partner/someone else' level. Table 2A has also been reformatted to mitigate against any confusion as a result of varying heights.

  1. Line 337: The word 'Girls' should not be capitalized.
  2. Table 2A: The reference to 'See graph' in page 14 is not informative enough. Also, the heigth of rows seems to change without a particular reason. Additionally, near the bottom of page 15 the 'hus/partner/someone else' reference seems to be repeated, with different values associated to it.

Response: Thank you. We have replaced ‘See graph’ with Figure 3a and Figure 7a respectively. The repeated factor level ‘hus/partner/someone else’ has been replaced with ‘With hus/partner/someone else’.

  • .The title of sub-section 4.4 should be reviewed.

Response: Thank you. Now reads as “2017 Senegal Demographic and Health Surveys (SDHS)”, see line 346.

  • In Figures 3 and 7, the legend is missing in the graphs on the right.

Response: Thank you. The legend is not missing. All three figures share the same legends in Figures (a) and (b).

  • Figure 5 seems to be displaced to the right and not fully visible.

Response: Thank you. Figure 5 is now in full view.

  • Lines 422, 429, 435 and 444: Perhaps 'had higher' should be written as 'had a higher'.

Response: Corrected, see line 428, 435, 442 and 449, respectively,

  • Line 426: Maybe 'continuations with POR of' could be written as 'continuation with POR values of'.

Response: Now reworded as ’Continuation with POR values of …, see, line 432.

  • Line 433: Perhaps 'terms household' could be written as 'terms of household'.

Response. Thanks. Omitted ‘of’ added (line 440).

  • Line 437: Maybe 'from Highest' should be written as 'from the Highest'.

Response: Thank you: Corrected, line 444

  • Line 439: Perhaps 'lower and lowest' could be written as 'Lower and Lowest'.

Response: Thank you. Corrected, line 446

  • Line 446: Maybe 'among girls' should be written as 'between girls'.

Response: Thank you. Corrected, line 453

  • Lines 447-448: Perhaps 'had formal job had lower' should be written as 'had a formal job had a lower', and 'had informal or' as 'had an informal one, or'.

Response: Thank you. Corrected, line 454

  • Lines 454-455 and 468-469: Maybe 'of social norms factor with respect to geographical' could be written as 'of the social norms factor with respect to the geographical'.

Response: Thank you. Corrected, line 461 & 475.

  • Line 456: Maybe 'lived Saint' could be written as 'lived in Saint'.

Response: Thank you. Corrected, line 463.

  • Lines 457, 458 and 467: Perhaps 'had significantly' could be written as 'had a significantly'.

Response: Thank you. Corrected, line 465, 474,

  • Line 470: Maybe 'to decreased' should be written as 'to a decreased'.

Response: Thank you. Corrected, line 477.

  • Line 473: Perhaps 'of proportion' could be written as 'of the proportion'.

Response: Thank you. Corrected, line 479

  • Line 477: Maybe 'had highest' should be written as 'had a higher'.

Response: Thank you. Corrected, line 484

  • Line 482: Please correct 'Tabe 2' to 'Table 2'.

Response: Thank you, Corrected, line 490

  • Line 485: Perhaps 'prevalence of' could be written as 'prevalence value of'.

Response: Thank you, Corrected, line 493

  • In Figure 8, the (a), (b) and (c) labels are missing.

Response: Thank you. Corrected, lines 498 – 499.

  • Figure 9 seems to be oversized.

Response: Thank you. Corrected, line 506 – 507.

  • Line 528: Please correct 'counterpatcs' to 'counterparts'.

Response: Thank you. Corrected, line 537.

  • Line 551: Maybe 'gave clearer' could be written as 'gave a clearer'.

Response: Thank you. Corrected, line 561.

  • Line 555: Perhaps 'that regions' could be written as 'those regions.'.

Response: Thank you. Corrected, line 565

  • Lines 556-567: The reference to the weakness in the study derived from it being reliant on the accuracy of the self-reportation of mothers, maybe affected by fear to prosecution, is very understandable and might compromise the relevance of its findings. Perhaps the results could be matched to those from other studies that would allow cross-checking in order to support the claims made, or else to include more objective complementary variables -perhaps coming from SHS as well- into the study to increase its strength.

Response: Thank you. This has been extensively reviewed to capture the intended message. Specifically, the section now reads “Although, the advanced Bayesian hierarchical geo-additive regression models employed here allowed us to accurately capture the potential spatial (geographical) structure of the prevalence of FGM/C among 0-14 years old girls in Senegal, particularly in relation to the key individual- and community-level drivers of the practice. We note, however, that there exist two limitations of the study we identified.  First, given that the  information on a girl’s FGM/C status were provided by their mothers, the issues of inaccurate self-reporting largely due to difficulty in recalling an event that took place several years ago or due to the fear of being prosecuted by the state under the provisions of the anti-FGM/C legislation in Senegal.  Secondly, we note that the datasets we analysed included information on girls aged 0-14 years only. Therefore, it is important to bear in mind while interpreting our results that the FGM/C status of a girl as at the time of the survey may not be her final status in that a girl who was not cut at age 14 may still be cut in the future. Nevertheless, it is highly unlikely that bias due to inaccurate reporting or recall bias will have a significant effect on the findings of this study. Besides, the statistical modelling approach employed in this study allowed the incorporation of multiple sources of variability in parameter estimation within a Bayesian hierarchical spatio-temporal regression modelling framework, thereby, allowing a straightforward quantification of uncertainties. ”

  • Line 578: Perhaps 'pressurized' could be written to 'pressured'.

Response: Thank you. Corrected, line 593

  • Line 579: Please correct 'exclusions.' to 'exclusion.'.

Response: Thank you. Corrected, line 594

  • Line 583: Maybe 'hotspots regions' could be written as 'hotspot regions'.

Response: Thank you. Corrected, line 598

  • Lines 585-586: The claim that 'Such interventions must involve religious, traditional and political leaders' maybe would require further elaboration to explain its intention and justification.

Response: Thank you. Steps have been taken to address the Reviewer’s points here and we have reviewed the entire paragraph to read “Social norm plays a significant role in the persistence of FGM/C in Senegal.  We found that a woman’s FGM/C status, her support for the continuation of FGM/C and her belief that FGM/C is a religious obligation, were key determinants of her daughter’s FGM/C status. Specifically, daughters of cut women, daughters of women who would not want FGM/C to be stopped and daughters of women who believed that FGM/C was required by religion, were most likely to experience FGM/C at some point in their lives.  There is a high likelihood of cutting a girl who lived in  a community where more than 30% of the female inhabitants were circumcised. In such a situation, girls are pressured by their peers to conform and get cut or risk social exclusion. Results show that FGM/C in Senegal varied temporally and spatially (geographically), peaked in 2005 with Matam, Tambacounda and Kolda identified as FGM/C hotspot regions  in Senegal, and also  identified Poular, Soninke, Diola and Mandingue ethnic groups as the FGM/C  high risk ethnic groups.

A tailored intervention should target the identified hotspot regions and ethnic groups in its design and implementation in partnership  with religious, traditional and political leaders who are generally revered within their communities and thus most likely to champion changes that would potentially bring about a total abandonment of the practice in Senegal”

  • Line 593: The reference to 'to reduce the risk in FGM/C outcomes.' is not clear, and perhaot could be rephrased.

Response: Thank you. See response to 45 above.

  • Line 598: Perhaps 'by Population' could be written as 'by the Population'.

Response: Thank you. Corrected, line 614

  • Line 608: This line seems to be incomplete.

Response: Thank you. Corrected, line 690.

REFERENCES: It is recommended to check the authors' names abbreviations, as they are shown using different formats on different references.

Response: Thank you. Reference listing now reviewed. Please see the References Section.

[1] Mackie, G.; LeJeune, J. Social Dynamics of Abandonment of Harmful Practices (2009).  A New Look at the Theory; Innocenti Working Paper No. 2009-06, Florence, UNICEF Innocenti Research Centre.

[2] Shell-Duncan, B.; Wander, K.; Hernlund, Y.; Moreau, A. (2011). Dynamics of change in the practice of female genital cutting in Senegambia:  Testing predictions of social convention theory. Social Science and Medicine,73, 1275-1283.

[3] Matanda, D. , Atilola, G., Moore, Z., Komba, P., Mavatikua, L., Nnanatu, C.C. and  Kandala, N-B (2019). Female Genital Mu-tilation/Cutting in Senegal: Is the practice Declining? In Press.